# Diagnostic and Therapeutic Value of Aptamers in Envenomation Cases

**DOI:** 10.3390/ijms21103565

**Published:** 2020-05-18

**Authors:** Steven Ascoët, Michel De Waard

**Affiliations:** 1L’institut du thorax, INSERM, CNRS, UNIV NANTES, F-44007 Nantes, France; ascoet.pro@gmail.com; 2LabEx Ion Channels, Science & Therapeutics, F-06560 Valbonne, France; 3Smartox Biotechnology, 6 rue des Platanes, F-38120 Saint-Egrève, France

**Keywords:** venoms, envenomation, toxins, diagnosis, aptamers, in vivo neutralization

## Abstract

It is now more than a century since Albert Calmette from the Institut Pasteur changed the world of envenomation by demonstrating that antibodies raised against animal venoms have the ability to treat human victims of previously fatal bites or stings. Moreover, the research initiated at that time effectively launched the discipline of toxicology, first leading to the search for toxic venom components, followed by the demonstration of venoms that also contained compounds of therapeutic value. Interest from pharmaceutical companies to treat envenomation is, however, declining, mainly for economic reasons, and hence, the World Health Organization has reclassified this public health issue to be a highest priority concern. While the production, storage, and safety of antivenom sera suffer from major inconveniences, alternative chemical and technological approaches to the problem of envenomation need to be considered that bypass the use of antibodies for toxin neutralization. Herein, we review an emerging strategy that relies on the use of aptamers and discuss how close—or otherwise—we are to finding a viable alternative to the use of antibodies for the therapy of human envenomation.

## 1. Introduction

The history of antivenom sera began with a French doctor, Albert Calmette, who, at the end of the 19th century, began immunization work using cobra venom with rabbits as animal models [1]. In this work, he described several immunization strategies; the most reputed, which is still in use today, was the administration of increasing venom concentrations in order to develop antitoxin antibodies. The novelty of this investigation was the fact that the serum of injected animals had both preventive and therapeutic properties, since injection of serum from an immunized rabbit to a control rabbit prevented the toxic signs of envenomation (for a complete description of the career endeavors of Albert Calmette, see [2]). Calmette’s protocol was thought to induce the production of highly specific immunoglobulin G that was considered protective against envenoming in vivo. It was rapidly demonstrated that the sera antitoxins combine directly with the venom toxins themselves to provoke in vivo toxicity neutralization. Next, it soon became clear that a given monovalent serum produced against the venom of a given snake species was of limited efficacy to neutralize the toxins from the venom of other snake species. Vital Brazil, who became the first director of the Butantan institute in São Paulo, developed the concept of polyvalent serum to enlarge its therapeutic value against the venoms from several species of snakes. In 1895, a cobra antivenom serum was produced in immunized horses and shown to have the proper characteristics for clinical use (potency and sufficient quantity) [1,3]. In 1926, the first serum made against Mexican scorpion envenomation was produced and endorsed by the Mexican federal health authorities for development [4]. In 1927, an antivenom serum was industrialized against rattlesnake poisonings for specific use in the United States (in a snake serum received at a city hospital to treat residents of Miami) [5]. In 1936, an antivenom was developed against black widow spider venom by Merck under the trade name Lyovac [6]. Australia, known for its numerous extremely venomous species, such as *Oxyuranus microlepidotus*, began producing its sera in 1930, with significant activity in the following years, producing no less than 11 antivenom sera directed against several animals, such as snakes, spiders, and jellyfish. Many technological improvements have occurred during the course of antivenom development, such as the pepsin enzymatic digestion to produce F(ab’)_2_ antibodies [7,8,9,10]; polyvalent antivenoms (such as that produced by Wyeth) [11]; affinity-purified Fab antivenom; the first venom-less antivenom, which was based on an immunization procedure against recombinant toxins by Bioclon [12]; or progress on bioengineered antivenoms that removes the need to use large animals for immunization [13]. For accurate reviews on the history of antivenoms, we refer readers to references [14,15,16].

## 2. Envenomation is still a Major Public Health Problem

Envenomation incidents have been classified as category A (the highest priority) of neglected tropical diseases by the World Health Organization (WHO) since June 2017 [17]. Among these incidents, it is understood that snakebites take the biggest toll with 1.8–2.7 million real cases of envenomation, and no less than 80,000 to 140,000 deaths annually, notwithstanding unreported cases in many rural areas with deficient health care facilities. Major clinical symptoms include breathing problems, kidney failure, tissue damage, and bleeding disorders. Because of the severity of the symptoms, it is currently believed to lead to 400,000 amputations globally, mainly because of tissue destruction, or other permanent disabilities. The burden is most severe in Southeast Asia, followed by Sub-Saharan Africa and Latin America.

## 3. Major Threats to Antivenom Production and Usage

Despite the technological advances in the production of antivenoms, shortages of supplies to treat clinical cases remain in countries where envenomation continues to be a critical issue [18,19,20]. This is due to several reasons. First, the legal requirements regarding the quality of manufacturing have grown over time, thereby increasing the costs of production. This is problematic for countries with a small market size, which additionally are characterized by a lack of (i) financial resources for their public health policies and (ii) regulatory authorities capable of controlling the adequacy and quality of antivenoms [21]. Some traditional antivenom producers have left the antivenom market (Behringerwerke AG, Sanofi Pasteur) [22]. Second, the use of antibodies that originate from horses, sheep, goats, or rabbits to treat envenomation is itself becoming a health concern. Due to their foreign nature, such antibodies can cause severe adverse reactions in humans upon injection, such as hypersensitivity responses or, worse, pyrogenic reactions and anaphylactic shocks. Several days after treatment (generally 8 to 12 days), patients may undergo serum sickness that is characterized by fever, allergic reactions, and cutaneous eruptions [23]. Third, in tropical areas, immunoglobulin G has a limited shelf-life, which further amplifies the supply issue problem. Finally, it should be mentioned that it is not rare that, within the complex venom mixture, many toxins escape the immunization process since they are non-immunogenic, while others are immune-suppressors [24]. The WHO has launched a roadmap that comprises several important aims: (i) evaluation of the safety and effectiveness of commercial antivenoms [25] and (ii) a two-fold reduction in mortality and disability by 2030. The WHO aims to facilitate the urgently needed collaboration among regulators, manufacturers, clinicians, health authorities, and national or international organizations [26]). The WHO action also comprises two interesting tools that can be consulted [27]. These include the “Guidelines for the production, control and regulation of snake antivenoms immunoglobulins” and the “Venomous snake distribution database” [28]. Combined, antivenom shortages and health-related issues with antivenom use in clinics should also prompt health researchers to find new alternatives to deal with poisoning as effectively as possible. Several alternatives are emerging, among which the use of aptamers appears to be promising.

## 4. Emerging Technological Alternatives to the Production and Use of Polyclonal Antibody-Based Antivenoms

The therapeutic area of antivenoms has witnessed interesting new technological developments. However, most do not yet offer a credible alternative to immunoglobulin G (IgG)-based antivenom marketing and sale. Below, we briefly summarize some of these technological developments (Table 1).

### 4.1. Toxin Activity-Based Antivenoms

Antivenoms have efficacies that are limited to the snake species whose venoms were used for antisera production. Hence, the expected paraspecificity of antivenoms—the capacity to neutralize toxins from venoms originating from phylogenetically distant species—is quite low. In one study, the paraspecificity of antivenoms was considered, not on the basis of phylogenetic or geographical distance of the species, but on the potential of these antivenoms to neutralize a class of toxin that is most relevant to the clinical manifestations of envenomation [29]. A clear demonstration of the efficacy of this approach was made by studying the neutralizing potential of antivenoms against procoagulant snake venoms for a large set of phylogenetically distant snakes whose venoms kill mainly by consumption coagulopathy. All of the antivenoms that passed this functional screening showed cross-reactivity by increasing the survival time of mice injected with venoms from those distant snakes. This medical-based approach, by selectively evaluating the potential of a wide-range of antivenoms on one of the main relevant clinical symptoms, allows for a fair prediction of the paraspecificity potential of a given antivenom.

### 4.2. Bioinformatic-Assisted Rationale Snake Antivenom Design

In many ways, the rationale described herein for this particular approach could be considered a follow-up of the antivenom selection method based on the clinical symptoms induced by the venoms themselves, as described above. The objective here was to apply a bioinformatics strategy for epitope predictions by focusing on venom metalloproteinases, since these enzymes are considered the most important targets to neutralize to avoid lethal hemorrhage in humans. By working on all isomers of this important class of enzymes within a venom, here, from the snake *Echis ocellatus*, seven epitopes with good immunogenic potential and present in all isomers were identified [30]. These authors demonstrated that the antiserum, produced against a single synthetic multiepitope (containing the seven epitopes) DNA immunogen contained antibodies active against several of the metalloproteinases of this venom and could cross-specifically neutralize the hemorrhage produced by several other snake venoms.

### 4.3. Monoclonal Antibodies

One avenue of research consists of producing monoclonal antibodies using the hybridoma technology, an approach first validated in 1988 for the in vivo neutralization of toxin II from *Androctonus australis hector* venom [31]. Since then, the concept has been corroborated with various monoclonal antibodies from murine hybridomas [32], but also for the production of human monoclonal antibodies using transgenic mice [33]. Another manner to discover monoclonal antibodies is based on the use of antibody phage display [34,35,36,37,38,39]. Additional efforts were made to produce monoclonal antibodies possessing cross-reactivity properties, i.e., that are capable of neutralizing two or more toxins that have similar primary structures. This approach resembles the polyvalent antivenom concept [38,66,67]. While these optimized monoclonal antibodies have undeniable advantages over polyclonal antibodies of foreign origin, the difficulty remains to produce as many monoclonal antibodies as there are toxic components, or at least life-threatening components, present within any given venom (from 20 to over 40 in some cases). Achieving this objective requires the precise identification of all toxic components within a given venom, according to the principle of toxicovenomics [68] that takes advantage of integrative venomics in understanding the pathological processes underlying snake envenoming [69]. An obvious limit of toxicovenomics is that a venomous compound shown to be toxic for a laboratory animal (generally murine models) is not necessarily also toxic in humans, which raises questions about the paradigms that should be applied to unequivocally identify lethal or health-hazardous toxins for humans. To date, the question of the feasibility of producing monoclonal antivenom cocktails for clinics has been answered only partially at this stage [39,70]. An estimate of the cost-effectiveness of this approach indicates that antivenoms based on the oligoclonal mixtures of human IgG antibodies would be in the range of USD60–250 per treatment, which is closely comparable to the current costs of polyclonal antivenoms [66]. Hence, efforts deployed in developing high-tech monoclonal antibodies should be supported by health authorities from countries in which snakebites are a major issue.

### 4.4. Other Technological and Chemical Initiatives

Among the other proposals that have been developed to neutralize toxic components, several original approaches have been published: nanoparticles [41], peptides, alternative binding proteins, [40,71], natural compounds from plants, and small molecule inhibitors [72,73,74,75,76,77,78,79]. The technological initiatives that bear the most resemblance to monoclonal antibodies are small non-antibody protein scaffolds that have the potential to be toxin binders thanks to in vitro selection technologies such as phage or ribosome display. Hence, the scaffolds that have emerged are named DARPins, Affibodies, Adnectins, Avimers, and Anticalins (for a review see [40]). These compounds have common properties and advantages over antibodies (smaller size, great stability, good half-lives, better tissue penetrance, low immunogenicity, kidney-mediated clearance, polyvalence if needed, lower cost of production, and easier chemical conjugations) [40]. Despite these advantages, their usage in neutralizing venom toxins in vivo remains to be validated. Several efforts have been undertaken to identify small molecules possessing activity against relevant venom toxins [42,43,80,81,82]. Varespladib, a broad-spectrum phospholipase A_2_ (PLA_2_) inhibitor, first developed to act on mammalian PLA_2_, was recently repositioned by Ophirex Inc. to inhibit snake PLA_2_ during envenomation [83]. The protective properties of Varespladib against envenomation appear highly promising, as it efficiently inhibited the hemorrhagic toxicity and muscle edema induced by *Deinagkistrodon acutus* and *Agkistrodon halys* venoms in vivo [44]. The drug showed potent inhibition of PLA_2_ activity of 28 medically important snake venoms with important survival benefits in vivo [43,80]. The idea emerges that a combination of this wide-spectrum PLA_2_ inhibitor with another large-spectrum metalloprotease inhibitor could represent an important therapeutic step forward for cases of snake envenomation. The finding that ethylenediaminetetraacetic acid (EDTA) efficiently inhibits zinc-dependent metalloproteinase and neutralizes snake venom-induced lethality in vivo [29] indicates that the association of Varespladib with EDTA could have interesting therapeutic value. 

The antivenom properties of plant extracts deserve attention, since plants represent the only viable alternative to modern medicine and pharmacology in Asia, Africa, and Central and South America. It is estimated that there are over 700 plant species that may display activity against snake venoms [45,46,47]. The wealth of literature on this topic has been compiled into a large accessible public phyto-antivenom database with different search options [48]. Various mechanisms of plant action have been invoked, such as: (i) venom inactivation by directly binding onto the toxins [49,50] or enzymes such as metalloproteases, hyaluronidases, and phospholipases A_2_ [51,52,53,54]; (ii) divalent metal ion chelation, with further action on metalloprotease and phospholipase A_2_ activities [55]; and (iii) antioxidant activities, by preventing tissue oxidative damage as a result of phospholipase A_2_ activity [54,56]. Mechanisms such as competitive block of target receptors, while conceptually interesting, are less likely to be an effective therapeutic solution at a large scale since modulation of the activity of these receptors contributes to a large extent to the global adverse effects at the clinical level. Nonetheless, these approaches may also be used in addition to traditional antibody-based antivenom administration to further improve clinical symptoms and vital outcomes of envenomation [67,84]. 

Another innovation of significant interest, as chief alternative to antibodies for toxin neutralization and detection, is the use of oligonucleotides [57,58,59,60,61,62,63,64,65]. Indeed, for several decades, DNA and/or RNA sequences, called aptamers, have been developed for various applications due to their antibody-like binding to a specific target [85,86,87,88,89,90,91,92,93,94,95,96,97,98,99,100,101]. It is only in recent years that the prospect of using them as new-generation antivenoms or as diagnostic tools has been considered. In the course of this review, the reasons why aptamers may represent viable alternatives to antibodies will be developed, some examples of use with regard to toxins and venoms will be provided, and the improvements that may be expected in the near future will be discussed.

## 5. Comparative Advantages in Using Aptamers over Antibodies

Perhaps the greatest advantage of aptamers over antibodies is the ease with which these chemical entities can be synthesized, while antibodies need to be produced either by animal immunization (providing finite amounts) or through the use of hybridomas or mammalian cell lines (requiring large culture facilities). Size-wise, they are much smaller than antibodies (between 12 and 30 kDa, compared to 150–180 kDa). Their production is easily reproducible and, compared to antibodies, cost-effective, which, combined with their long storage life (several months or years) and temperature-insensitivity, means problems with shortages of supply are unlikely. For instance, for a company specialized in aptamer development, Aptagen, the cost of aptamers would be six times lower than antibodies per unit weight (50 USD/g versus 300 USD/g, although these costs are expected to lower dramatically as a function of the scale of production [66,102,103]). Closer to clinics, a comparative cost-effectiveness of therapy for age-related macular degeneration has been performed between Verteporfin (a ribonucleic acid aptamer targeting the vascular endothelium growth factor (VEGF) from Novartis Pharma AG) and Ranibizumab (a monoclonal antibody targeting a wider diversity of VEGF isoforms from Genentech Inc). It was found that the treatment cost with the aptamer was half that of the monoclonal antibody [104]. The difference is not as striking as could be expected, but an emerging consensus suggests that the costs should be slightly lower. The possibility to add conserved nucleotide sequences at their 5′ and 3′ ends also allows for bulk amplification of mixtures of aptamers directed towards a large diversity of toxins. Because they are synthetic products, the production process is less inclined to lead to bacterial or viral contamination. Though this is conceptually not intuitive, aptamers can be target-selective as demonstrated by the fact that they can distinguish between theophylline and caffeine, which differ by a methyl group, with a 4-log difference in affinity. The conditions of preservation of activity (room temperature versus freezer for antibodies) is another competitive advantage [102]. For in vivo applications [92,101], it was reported that aptamers are poorly immunogenic, lack toxicity, and diffuse fast and readily into organ tissues, a property that should be ideal for toxin neutralization. In comparison, for antibodies to neutralize toxins deeper in tissues, Fab-based antivenoms have been used, but at the expense of reduced serum half-life [66]. A distinctive advantage of aptamers is their ability to bind to a wide variety of targets, from small molecules to cells or nano-objects. Therefore, this property should be useful for the identification of aptamers able to interact with venom components that differ by size and/or chemical nature (transmitters, alkaloids, peptides, proteins). It is expected that a fraction of these aptamers should possess toxic neutralizing capabilities. The most promising property of aptamers is probably the infinite possibilities of chemical engineering thanks to the orthogonal attachment of new chemical entities during synthesis, a property that it shares with other small non-antibody protein scaffolds such as DARPins, but that are more difficult with antibodies [105]. Aptamers have been tagged with fluorophores [106] or active pharmaceuticals, allowing for a multiplicity of applications: bio-imaging [98,107,108,109,110,111]; agent delivery [107,112,113]; diagnostic uses [98,106,114,115,116]; therapeutic uses [87,117,118,119], mainly for cancers [120,121,122], including the first aptamer on the market against VEGF [123]; theragnostic uses [124]; nanotechnology [106,112]; and bio-sensors [122,125,126,127] for the detection of pharmaceutical residues in the environment [128], codeine [129,130,131], or even anthracyclines [132]. The success of aptamers is illustrated by those aptamers that reached the market or are engaged in clinical trials [94,97,133,134,135,136,137]. We previously noted the case of Verteporfin for age-related macular degeneration. In 2010, several others were in clinical trial: AS1411 from Antisoma against nucleolin (phase II, acute myeloid leukemia); REG1/RB006 from Regado Biosciences against nucleolin (phase II, percutaneous coronary intervention); or ARC1779 from Archemix against A1 domain of von Willebrand factor (phase II, thrombotic microangiopathies) (for more complete descriptions of aptamers in clinical trials, see [92]). 

There are several drawbacks that deserve to be mentioned about aptamers compared to antibodies. The most limiting of these is probably the broad ownership that the company Archemix possesses on intellectual property (IP) rights for any discovered aptamer binding onto proteins [103]. How licensing and IP rights have evolved over time deserves a thorough investigation. The in vivo half-life of aptamers is much lower (~20 min) than antibodies (days to weeks) [85]. Exceptionally, a half-life of 24 h was reached for an aptamer on one occasion [103]. However, this rapid clearance rate from circulation, which is due to degradation by nucleases and their small size, can be prevented by polyethylene glycol [134] or cholesterol conjugation for improved circulation time, but also by using modified nucleotides with different base, sugar, or inter-nucleotide linkage chemistry that prevents nuclease degradation [92,105,138]. The third limitation of aptamers is their evident inability to reach intracellular targets, thereby limiting the scope of certain applications. The fourth limitation of the use of aptamers is perhaps the affinity for their targets. This remains a disputed claim, however, since dissociation constants that have been observed ranging from the µM to the nM level, with exceptional cases at the pM level [92,102]. For an interesting comparison between aptamers and antibodies, see references [85,99,105].

Although aptamers possess evident advantages, mandatory procedures need to be respected for the selection of aptamers that bind and hopefully prevent the toxic effects of venom components. 

## 6. Aptamer Selection Procedures and Property Improvements 

Systematic Evolution of Ligands by Exponential enrichment (SELEX) is an in vitro strategy for the selection of aptamers that is based on a library of randomly synthesized oligonucleotides (Figure 1). This technique was first described by Gold and Szostak in 1990, while they aimed at producing a high-affinity and specific bacteriophage T4 DNA polymerase RNA ligand [91,139]. A selection round by SELEX takes place as follows: (i) synthesis of the oligonucleotide library that contains both a constant sequence for amplification and a random sequence for screening; (ii) interaction assay of the oligonucleotide library with the natural target; (iii) extraction of the complex formed by various possible techniques such as a column, magnetic beads, or electrophoresis; (iv) dissociation of the oligonucleotide-protein complex; and (v) sequence amplification of the oligonucleotides by PCR or RT-PCR (first selection round). These steps are repeated until refined oligonucleotide sequences are obtained (five to 20 selection rounds depending on the nature of the targets). This method is the “classic strategy” for developing aptamers, which can be either RNAs or DNAs. Over time, this method has been modified and optimized to reduce the time required to achieve success but also to gain in affinity. In 1998, an automated-SELEX procedure was described [88]. To save time in selecting aptamers, Berezovski et al. moved away from the PCR amplification step by developing the non-SELEX method. This method saved an impressive amount of time since the aptamer selection process takes place in one hour instead of several weeks [105,140,141]. Recently, using this new method, the “ERaptR4” aptamer was developed in silico for the diagnosis of breast cancer. This aptamer recognizes the α receptors of estrogens that are overexpressed in this type of cancer [142]. To ensure that aptamers perfectly recognize the native conformation of proteins expressed at the cell surface, a cell-SELEX version of the selection method was also developed that uses whole cells to identify new aptamers [143]. To be efficient, this method requires a counter screening with negative control cells not expressing the markers of interest. In recent years, several aptamers have been developed for various cell targets using this method: nasopharyngeal carcinoma biomarker [144], renal cell carcinoma [120], and metastatic colorectal carcinoma [121]. Among other developments of interest, it is fair to mention the split oligonucleotide synthesis methodology that allows the production of oligonucleotide libraries (with unmodified or modified nucleic acids), in which single oligonucleotide sequences are attached covalently onto single beads [145]. The advantage of this technology is the ultra-fast identification (without multiple rounds of selection) of protein-binding aptamers and the fact that it is compatible with phosphorothionate or phosphorodithionate oligonucleotides that possess higher affinity for protein targets, as well as nuclease resistance properties [146,147,148,149]. For an update on aptamer selection technologies, see reference [85].

As mentioned earlier, several chemical optimizations have been developed to tackle the issue of aptamer stability in biological media, a prerequisite for therapeutic applications [85,105]. These chemical modifications have an obvious cost that needs to be weighed against the benefit they provide. For instance, in the case of venom neutralization, optimization of 20–100 aptamers may be considered a threat to their development for this application. The ambition of these aptamer chemical modifications is to make them more stable against enzymes such as exo- and endonucleases [89,134] that are naturally present in biological environments. Exclusively performing chemical modification at the 5′ or 3′ ends of aptamers is not 100% effective and may occasionally reduce their affinity for their targets. A new family of aptamers has emerged: the spiegelmers^®^ (Figure 2a). Basically, these are L-RNAs that correspond to the mirror image of a D-RNA (classic aptamer). The advantage of spiegelmers^®^ is their intrinsic resistance to degradation since nucleases, which are stereospecific of nucleotides, will not recognize or degrade them. The stability of spiegelmers^®^ in a biological environment was demonstrated by Klussmann et al. in 1996 [150]. However, to identify active spiegelmers^®^, the SELEX method has to be adapted using a mirror of the natural target. In short, the target itself should be an enantiomer of the natural protein, for instance (built with D-amino acids rather than with L-amino acids), and the SELEX procedure is started with classical aptamer sequences. Once interesting aptamers are selected, they are then synthesized in the L-configuration, making them de facto spiegelmers^®^ active against the biologically active configuration of the protein with its L-amino acids [151]. Obviously, spiegelmers^®^ can be produced only if enantiomers of the natural targets are also produced. This method is therefore not applicable to whole venom, but only to selected peptides or proteins that can be reproduced by chemical synthesis. For the applicability to venom toxins, examples have been provided in the past in which complex peptide stereoisomers (with several disulphide bridges) can be chemically synthesized [152,153,154]. Therefore, because of the difficulties associated with the chemical synthesis of toxin enantiomers, and the greater number of steps involved in aptamer selection and synthesis, it is obvious that this approach should be limited to the neutralization of a given toxin in vivo (and not toxin mixtures). Among other possibilities, spiegelmers^®^ might be used to perform snakebite diagnostics if they possess selectivity properties against a toxin that best exemplifies a given snake species. Alternatively, the use of spiegelmers^®^ could be envisioned for spiking traditional antivenoms if they are developed against some of the most toxic venom components. Several spiegelmers^®^ are currently in clinical study, such as NOX-H94 (which was the first that entered clinical trials) against hepcidine in the treatment of chronic anemia, the first clinical L-RNA [134,135]. Another spiegelmer^®^ in clinical trials is NOX-A12, an SDF-1/CXCL12 inhibitor to prevent an interaction between hematopoietic cells and bone marrow, and improving anti-tumor treatments [133,134,137]. The complete history and application of spiegelmers^®^ has been presented recently in a review [155]. In addition to spiegelmers^®^, there are also SOMAmers that incorporate modified nucleotide bases (Figure 2b). The aim is to make aptamers also compatible for the interaction with difficult-to-access sites when dealing with classical aptamers. For instance, one objective could be to deal with hydrophobic interactions that may be important for toxin activity. These types of interactions are difficult to access when using aptamers developed with classical nucleotides by increasing their affinity and specificity. The modified aptamers would therefore contain uridine nucleotide analogues with a large hydrophobic substituent on the nucleic base (i.e., carbon position 5 of uridine) [156]. A collateral advantage of the use of SOMAmers is that these substitutions greatly increase the half-lives of the aptamers in vivo. In the same vein of chemical engineering of oligonucleotides, a new generation of hybrid aptamers was developed recently, called X-aptamers, that combine the use of monothiophosphate backbone-substituted aptamers (for improved stability and affinity for proteins) and the incorporation of chemically-modified uridine (called X) to allow the addition of drug-like compounds onto the aptamers or new functionalities [96,157]. This technology is compatible with the pseudorandom bead-based aptamer libraries for fast track aptamer selections against any protein target [145,158]. Several aptamers have already been developed in this vein for the detection of proteins in schizophrenic patients [159] or as new forms of cancer treatment [160]. The attractiveness of aptamers is best illustrated by their ever-increasing number in various phases of clinic evaluation (see the reviews in references [97,100,161]).

Among the other chemical modifications that are worth mentioning, there are “escort aptamers” [93] and multivalent aptamers (Figure 2c,d). Escort aptamers are in principle aptamers modified to carry a new chemical functionality, such as a fluorescence tag. An escort aptamer therefore becomes a valuable imaging tool that may replace an antibody if the selectivity is high enough for the target. Recently, an escort aptamer was developed to improve the affinity of melittin (from bee venom) to treat cancer [113]. The properties of an aptamer can also be enlarged to new functionalities in addition to imaging to lead to a theragnostic agent. Such a theragnostic tool was developed by coupling an aptamer to a nanoparticle for imaging, that itself served as a platform to immobilize doxorubicin, an anthracycline to treat solid tumors [86,162]. Multivalent aptamers are, as indicated by their name, aptamers that duplicate the same function (recognition of a ligand, for instance) or that combine two different aptamers recognizing different targets, themselves separated by a neutral linker [107,163,164]. Multivalent aptamers, like more classical aptamers, can obviously also be optimized or work as escort aptamers [93,165]. These examples all illustrate the wonderful adaptability of aptamers to various engineering approaches.

## 7. Use of Aptamers as Antitoxins

For most applications, aptamers seem to have valuable properties for diagnostic and/or therapeutic purposes. Hence, they are envisioned as tools to detect specific markers (such as on cancer cells) or as ligands to induce an appropriate pharmacological response. For therapeutic applications, aptamers are mainly used for the treatment of cancers, in particular because their specificity is supposedly equivalent or superior to that of current anticancer agents [94]. In none of these applications did researchers consider aptamers for their abilities to bind and neutralize blood circulating components in vivo. At best, they have been used for the detection of doping or polluting products in water. Yet, we have seen that aptamers can be designed for high affinity, high selectivity, and nuclease resistance for in vivo applications. Aptamers have interesting properties for interacting with proteins (most venoms posing a medical threat contain peptides and proteins). Indeed, the recognition of proteins by nucleic acid sequences involves nucleic acid bases and phosphate ester backbone interactions with side chains and a peptide backbone [166,167,168]. As part of this review, we will thus focus on the emerging application of aptamers concerning toxin detection and neutralization, or for the treatment of envenomation. We develop herein several examples that illustrate the valuable potential of aptamers in animal toxinology. All the aptamers described here are summarized in Table 2 (with information such as target, sequence, and K_D_).

### 7.1. DNA Aptamers Directed against Bungarotoxin from Venom of the Elapid Bungarus Multicinctus

The genus *Bungarus* is widespread throughout Asia (except the Philippines) and the *multicinctus* species is concentrated in Southeast Asia and Taiwan [61]. The venom of *Bungarus multicinctus* is composed of a cocktail of neurotoxins: α-, β-, γ-, κ-bungarotoxin acting mainly at synaptic levels [65]. Although the bite of this species represents only 7.5% of all snakebites in Taiwan, it is considered one of the most dangerous bites, with a 7% to 50% lethality occurrence if timely and appropriate antivenom treatment is not administered. Specifically, this venom induces paralysis and breathing difficulties in the victim before death occurs. Antivenom drugs are being developed against the majority of local venomous snakes and it is only recently that a study of patient management has been established [169]. In order to further improve the treatment of these bites, aptamers have been developed to neutralize some components of the venom of *Bungarus multicinctus* [61,65]. Historically, the first aptamer was produced against α-bungarotoxin. Incidentally, this was the first oligonucleotide ever developed as being active against a venom component. The four disulphide-bridged and three-finger toxin α-bungarotoxin, with its 61% abundance, is the main component of the venom of this snake [65,170]. This 74-amino acid toxin is an irreversible competitive antagonist of the nicotinic acetylcholine receptor of the neuromuscular junction. Hence, it has been largely used for the molecular identification and biophysical and pharmacological characterization of this receptor [171]. The aptamers against α-bungarotoxin were identified by conjugating a fluorophore onto a DNA oligonucleotide library and screening towards aptamer/toxin complex formation using toxin immobilization on a solid surface. Unbound aptamers were washed out. The aptamer sequences were then subcloned into an *Escherichia coli* (*E. coli*) strain for the purpose of sequencing. Several aptamer sequences could be identified according to this procedure, but only two (clones 24 and 51) displayed a high affinity for α-bungarotoxin. Observed K_D_ values were in the upper micromolar range, which is currently considered high for interacting aptamers [61,65]. In 2014, a team decided to develop an additional aptamer, but directed specifically against β-bungarotoxin, another toxin of the venom of *Bungarus multicinctus*. This toxin is less abundant than α-bungarotoxin in this snake venom, but it has a higher lethality potential, making it an interesting target for an improved treatment of envenomation. Using the SELEX platform method, up to four aptamers were developed (βB-1, βB-20, βB-19, βB-32), with each possessing an affinity in the range of 10^-8^ M, which is a greater guarantee for their specificity in targeting this toxin. The research also demonstrated that one of these aptamers, βB-1, is in fact specific for recognizing a component of the *Bungarus multicinctus* venom since no affinity was detected for venom components of different species [65]. This finding suggests that aptamers may be used to diagnose the species of snake that has bitten a human victim based on the identification of a circulating toxin type. However, while this property may appear desirable for diagnosis, it is a less encouraging characteristic if the aptamer under development is planned for integration into an aptamer cocktail with polyvalent interacting and neutralizing specificities. The two following examples illustrate the potential level of polyvalence that can be reached by aptamers.

### 7.2. Engineering of DNA Aptamers as an Approach to Identify New Ligands from the Indian Bungarus Caeruleus (Krait) Snake Venom

One of the major problems with snakebites remains the identification of the snake species in order to administer the right treatment. Other than the Seqirus snake venom kit, whose reliability has been questioned [172], it is fair to state that there is currently no precise diagnostic test to identify the nature and origin of the venoms. However, it remains an important issue to tackle because venom composition differs between two species of venomous animals, even if they are related. Dhiman et al. proposed using the aptamer targeting α-bungarotoxin [61], here called α-Tox-FL, to identify new aptamers to structurally related toxins. The idea was to develop aptamers capable of recognizing toxins present in the venom of *Bungarus caeruleus*, known to contain an α-neurotoxin that possesses 80% sequence homology with α-bungarotoxin [58]. Using bioinformatics tools, they studied the secondary structures of the aptamer directed against α-bungarotoxin in order to truncate it into two variants: α-Tox-T1 (14 nucleotides) and α-Tox-T2 (26 nucleotides). The affinities of these aptamer variants for the raw *Bungarus caeruleus* venom were investigated using an Enzyme Linked Aptamer Assay (ELAA). The data showed that the α-Tox-T2 variant had the best affinity for the venom with a K_D_ of 2.8 nM compared to the second variant (K_D_ = 44.8 nM) of lower affinity and the parent aptamer (K_D_ = 18.0 nM). Interestingly, the parent aptamer and the α-Tox-T2 variant were shown to be selective for raw *Bungarus caeruleus* venom. This led the authors to conclude that these oligonucleotides were specific for the α-toxin present in the crude venom [58]. However, this demonstration was incomplete since they did not use the pure toxin in isolation, but instead used the whole venom with its complex mixture. By using the crude venom for this study, these authors cannot prevent the possibility that these aptamers bind onto a venom component different from the α-toxin. Nevertheless, this study demonstrates that the aptamer binding onto a component of *Bungarus caeruleus* venom is independent of the species’ geographical origins. This study, therefore, also highlights the value of using aptamers for the development of new antivenom drugs, although it is likely that finding one aptamer capable of binding onto its target, regardless of the geographical origin of the venom, is not a guarantee that all aptamers possess such a property. The same also holds true for antibodies: some have cross-geographical capabilities, while others do not possess this quality. The value of this work remains the illustration that aptamers are likely to possess cross-reactivity properties for toxins of similar folds and a reasonable amount of sequence homology. This raises an interesting question about the relative intrinsic capabilities of aptamers and antibodies to best recognize a family of related toxins. Intuitively, single aptamers may be compared to monoclonal antibodies. Considering the physicochemical properties of aptamer/toxin interaction sites, which should be based on a larger chemical space than the antibody/toxin interaction, we may assume that aptamers have better intrinsic polyvalence properties than antibodies. Such an assumption obviously awaits a formal demonstration. However, it is interesting to note that in 2001 an alternative SELEX method called “toggle” SELEX, was developed to isolate cross-reactive aptamers using several targets of similar nature during selection [173]. Such a method, combined with the power of venomics and identification of most toxic venom compounds, would have the potential to boost the development of cross-reactive aptamers. There is no equivalent existing technique for the facilitated selection of cross-reactive antibodies [99]. 

### 7.3. DNA Aptamers against Cardiotoxin from Naja Atra Snake Venom

Several studies have shown that aptamers are capable of cross-reactivity, i.e., although the aptamer is specific for a ligand, it may possess a good affinity for a homologous toxin [173,174]. Snake venom cardiotoxins are toxins with the classical three-finger pattern, a pattern that is also present in the α-bungarotoxin neurotoxin. In addition, sequence alignments have shown some sequence homology between cardiotoxins and α-bungarotoxin, as well as a preserved disposition of cysteine residues within the sequence [57]. Based on these two premises, the researchers studied the affinity of specific α-bungarotoxin aptamers for the *Naja atra* cardiotoxins. In order to measure the affinity of these aptamers for the cardiotoxins, these aptamers were modified in such a way that they contained a fluorophore at their 5’ ends and a fluorescence quencher at their 3’ ends. Interaction of these aptamers with the cardiotoxins resulted in a decrease of the fluorescence level. Aptamers previously developed against α-bungarotoxin [61] are here named bgt1, bgt2, bgt3, and bgt4. Initially tested on cardiotoxin 3 (CTX3), the aptamers showed an affinity in the range of 10^-8^ M, with the exception of bgt1. The study was further developed by studying the affinity of bgt1 on homologous CTX3 cardiotoxins: CTX1, CTX2, CTX4, CTX5, CTXN, and CLBP [57]. This pioneering study perfectly illustrates the level of cross-reactivity that aptamers intrinsically possess for three-fingered toxins from snake venoms. They undoubtedly are precursor investigations for the future development of polyvalent antivenom aptamer cocktails that have the potential to neutralize three-fingered toxins during envenomation. While the development of these aptamers clearly illustrates the capability of these entities to bind toxins in vitro, most of these studies formally lacked the demonstration that aptamers represent valuable tools as substitutes to antibodies for in vivo neutralization of toxins and the saving of lives. The following example illustrates this point.

### 7.4. Neutralization of a Lethal Venom Toxin, αC-Conotoxin PrXA, In Vivo by a DNA Aptamer

α-conotoxins are notoriously dangerous to humans because they are fast killer compounds through their paralytic action (1 mg of an α-conotoxin would be enough to kill a human, and 1 kg enough to kill 1 million people) [175]. They are some of the rare animal toxins of venom origin listed as potential bioweapons by the USA Center for Disease Control and Prevention. They were included in 2002 in response to the Public Health Security and Bioterrorism Preparedness and Response Act. However, despite the threat they represent, there is to date no working antidote against these peptides. While cases of poisoning by marine cones are far less frequent than snakebites, cone toxins remain an interesting avenue of investigation due to their lethality potential. Many are of smaller size and compact structure due to disulphide bridges, and orally available analogues have appeared, using head-tail cyclisation [176], further enhancing the risks of misuse. For a proof of concept of the utility of aptamers, one such conotoxin was chemically synthesized with high yield in its biologically active conformer with a single disulphide bridge. αC-conotoxin PrXA was originally identified in the venom of *Conus parius* and shown to be a potent nicotinic acetylcholine receptor antagonist [177]. In a mouse phrenic-hemidiaphragm nerve-muscle preparation, low concentrations of αC-conotoxin PrXA (IC_50_ close to 20 nM) lead rapidly to inhibition of twitch contraction of the muscle by blocking ACh-mediated activation of postsynaptic acetylcholine receptors [59]. Less than 15 min are needed to produce a full block of this muscle type in vitro, and therefore, the expectations would be that the toxin effectively kills mice by respiratory failure. Subcutaneous or intraperitoneal administration of 9 µg or less to mice is enough of a quantity to paralyze the animals and to induce death with 5 min latency. Longer latencies to death were observed for lower concentrations. Nevertheless, these observations illustrate that the toxin is efficiently directed to its site of action and blocks diaphragm muscles. In vivo, the kinetics of blocking are surprisingly faster than in in vitro experiments, indicating that full blockage of muscles is perhaps not needed for lethality. As such, this toxin represented an interesting lead compound to test, for the first time, the in vivo efficacy of aptamers in neutralizing the lethal effect of this toxin. The rapid mode of action of the toxin was an interesting challenge to overcome because neutralizing such a fast-acting toxin by aptamers would mean that the mode of action of aptamers in vivo is no less fast. Moreover, working on a reduced time scale at the experimental level had the advantage to avoid potential problems linked to the chemical stability and pharmacokinetics of classical aptamers. This bias obviously comes at the expense of a more complex situation of real envenomation, where numerous toxins may remain active for periods exceeding 24 h. However, we have seen that there are solutions to solve such a problem, should it arise, by making more in vivo stable aptamer analogues. To select for aptamers capable of interacting with αC-conotoxin PrXA, a SELEX procedure was launched that uses capillary electrophoresis and compound separation by charge. This type of separation performs well in the case of animal toxins because a large majority of toxins have a positive net global charge, whereas the opposite is true for aptamers. This is, however, not a general reality as a significant number of acidic peptides/proteins have also been observed in an Elapidae snake venom [178], suggesting that other modes of collection may be needed for snake venom components. In the case of the aptamer/αC-conotoxin PrXA complexes, they could be collected within an elution window located between the elution of the peptide and that of the aptamer library. This observation deserves a comment: the generally observed charge complementarity observed between aptamers and toxins indicates that one of the most likely chemical rules of interaction will be electrostatic. Several rounds of SELEX procedure were performed (four in total) and led to the identification of at least 10 aptamers after sequencing that could be classified into three distinct groups of homologous sequences. Due to the synthesis of a fluorescent analogue of αC-conotoxin PrXA, the affinity of these DNA aptamers for the toxin could be measured by fluorescence anisotropy change and were shown to range between 120 nM and 5 µM. The rather low affinity of these aptamers is presumably linked to the design of the selection procedure that relied on high peptide concentration and a limited number of rounds of selections whose stringency were not augmented over time. There is thus obvious room for improvement in selecting higher affinity aptamers. Structural predictions of the aptamers tested demonstrate a surprisingly high level of diversity, further indicating that there are not predictable rules of aptamer/toxin interactions. The aptamers were then tested for their efficacy in preventing αC-conotoxin PrXA binding onto the acetylcholine receptor. αC-conotoxin PrXA possesses the ability to prevent the binding of an iodinated analogue of α-bungarotoxin on the acetylcholine receptor. It was found that this inhibition could be prevented by the preincubation of αC-conotoxin PrXA with one of the selected aptamers with an IC_50_ value that was quite compatible with the K_D_ of interaction of this aptamer with the toxin. This first result was an indication that the pharmacophore of αC-conotoxin PrXA could be masked for its interaction with the receptor. Not all aptamers reacted this way, further illustrating that they do not fully “dress up” the toxin, like a molecular coat, a matter of interest considering that aptamers were relatively larger (77 nucleotides) than the peptide itself in its compact folded conformation (32 amino acids). The same active aptamer was also efficient in preventing the αC-conotoxin PrXA-mediated inhibition of twitch contraction, raising considerable hope that it may also be active in vivo. Of interest was the fact that in vitro an inactive aptamer (on the α-bungarotoxin test) was also not capable of preventing the toxin-induced inhibition of the mouse hemidiaphragm muscle contraction. This observation indicates that non-neutralizing aptamers that bind to the toxin do not prevent them from accessing the receptor in the tissue context by accessibility alteration. An in-depth analysis of the mode of action of the aptamers demonstrated that some aptamers shift the affinity of αC-conotoxin PrXA for their receptors, thereby potentially decreasing their efficacy [60]. The most remarkable results were obtained in vivo. At low concentrations of aptamers, death occurrence in mice could be considerably delayed (up to 80 min instead of less than 5 min), suggesting that toxin neutralization by aptamers had an operational window duration of at least 80 min. Beyond this duration, all mice could be saved, indicating that sufficient toxin clearance/degradation had occurred within this time frame. The fact that lethality could still be observed within this 80-min period may suggest a faster degradation kinetic of the aptamer in vivo than that of αC-conotoxin PrXA, and possibly regain lethal activity, while the aptamer is progressively degraded/eliminated with time. Part of the increase in latency in death occurrence comes from the fact that a fraction of the toxin is probably not neutralized, but it was observed that, in the absence of aptamers, the longest delay in death occurrence was 25 min with low toxin concentrations, thus, below the 80 min observed in the presence of the aptamer and the highest concentration of αC-conotoxin PrXA. At higher concentrations of aptamers (about 4–5 µg/mouse), a complete inhibition of mice lethality was observed, indicating that the toxin no longer could induce paralysis. The mice were observed for 24 h, presumably much longer than the aptamer stability in vivo, and there were no additional signs of toxicity due to αC-conotoxin PrXA being liberated from the toxin/aptamer complexes in circulation. This finding is a sign that enough of the toxin had been neutralized, metabolized, or eliminated by kidney filtration. In several experiments, the neutralizing aptamer was efficiently preventing mouse death when it was co-injected with the toxin. The fact that it was working in these conditions initially demonstrated that the complex does not dissociate in circulation in vivo [59]. One fear was that aptamers would lack selectivity and start interacting with all kinds of protein partners thereby “losing interest” in the toxin against which they were initially selected. More interesting was the fact that this aptamer also worked if it was injected after the toxin started to paralyze the mouse (i.e., rescue mode). The fact that the locus of injection was totally different was also proof that the aptamer diffuses very rapidly in the body and tissues, and easily finds its toxin partner despite a complex molecular and cellular environment. Here again, the finding was that the aptamer was highly selective for its peptide partner and behaved as a warhead for the neutralization of the lethal toxin. Finally was the interesting finding that an interacting aptamer, unable to prevent the binding of αC-conotoxin PrXA on its receptor in vitro, was also working efficiently in vivo to neutralize the lethal action of the toxin [60]. This effect must be mediated by an alteration of the normal biodistribution of the toxin. All of these observations hint at the importance of better examination of the pharmacokinetics of αC-conotoxin PrXA, the aptamer alone, and the αC-conotoxin PrXA/aptamer complex in vivo with appropriate imaging tags. Whatever the conclusions of these additional future studies, the present data themselves are a formal demonstration that aptamers can work for the in vivo neutralization of deadly toxins, and that they can act by two different mechanisms: (i) by directly covering, literally like a molecular coat, the pharmacophore of the toxin, which is the best mode of action; and (ii) by altering the normal biodistribution of the toxin that normally “aims” at targeting its receptor. Most likely, antibodies act by the same mechanisms. 

What are the improvements that may be expected in the future regarding this study? First, it is likely that there is room for upgrading the quality of the SELEX procedure in order to identify still higher affinity interacting aptamers. The aptamers that were tested are of reasonably high affinity, but much higher affinity aptamers are possible by a factor of 10–100. It would be interesting to correlate the affinity of an aptamer with its neutralizing potential in vivo. Second, the finding that aptamers have greater value in vivo than in vitro (generally it is the opposite) raises questions about the pertinence of using crude aptamer mixtures that target the same toxin. We would expect such a mixture to be more efficient in functionally neutralizing the toxin (i.e., several molecular blankets entirely covering the toxin). Third, while the data are convincing on neutralizing a fast-acting toxin, a similar investigation should be repeated on toxins possessing slow lethal activities. The issue will be to test whether the half-life of the aptamers in vivo is long enough to maintain their neutralizing potential. Intuitively, however, if the aptamers have the potential to neutralize αC-conotoxin PrXA, a fast-acting toxin, and degrade rapidly, then the toxin might regain activity within the 24-h frame of observation. This was not observed, suggesting that aptamers bound to the toxin synchronize their fate of elimination with that of the peptide. The obvious alternative interpretation is that αC-conotoxin PrXA is less stable than the aptamers, although most studies on toxins and aptamers conclude that aptamers should be less stable. Clearly, these issues need to be examined via pharmacokinetic investigations to provide definitive conclusions. Fourth, it would be of obvious interest to determine how much better aptamers, optimized for longer half-lives in vivo, would behave regarding the target to neutralize. In this respect, the next formal demonstration of the efficacy of aptamers in toxicity neutralization should occur with toxins that have long half lives in vivo and slow clearance rates. Finally, the ultimate goal, and next significant challenge, is to produce an aptamer mixture against full venom and assess the neutralizing potential in vivo of this cocktail of aptamers. It is fair to mention that a formal demonstration of an aptamer cocktail that is capable of clinically neutralizing a full venom, at the present time, remains elusive. At the current time, we can only guess at the steps required to achieve such an objective. The ultimate approach probably involves finding a rapid “single pot” technology to isolate hundreds of aptamers directed towards no less than a hundred different venom toxins. The proof of concept would gain value if it were performed against a snake venom of public health concern. The opinion of these authors is that it is technologically realistic. Getting to this point would definitively set the trend for future efforts to find alternatives to serum antibodies. Thus, the obvious subsequent challenges would be (i) producing sufficient quantities of these aptamer cocktails, (ii) ensuring that they can be produced reproducibly in similar relative ratios, (iii) ensuring their absence of toxicity in vivo in humans, and (iv) comparing their advantages to the gold-standard antiserum.

One of the issues addressed in the El-Aziz study was the specificity of the aptamers. The αC-conotoxin PrXA-blocking aptamer was inefficient in counteracting the effects of waglerin on muscle contraction. The effect of this snake peptide of 22 amino acid residues, which also contains a single disulphide bridge, on muscle contraction was not affected by the aptamer. However, αC-conotoxin PrXA and waglerin differ quite drastically in terms of sequence homologies, and it would be of interest to investigate the neutralizing potential of the working aptamers on several other α-conotoxins, of different species origin, that share greater homology with αC-conotoxin PrXA and that act on the same receptor. This would allow evaluating the polyvalence potential of a given aptamer and define the cut-off sequence homology whereby an aptamer no longer stays active. 

### 7.5. First RNA Aptamers Targeted against Loxosceles Laeta Spider Toxins

The genus *Loxosceles* is found in America and its venom contains harmful toxins such as sphingomyelinase D (SMD) isoforms. Envenomation by this spider can cause severe loxoscelism (skin necrosis). The *L. laeta* species, found in South America, is most commonly responsible for cases of envenomation. Many antivenom drugs are available on the market or in pre-clinical studies to treat these cases of envenomation. Nevertheless, they have many limits common to antibodies used for this purpose [64]. Thus, Sapag et al. selected specific aptamers of recombinant sphingomyelinase D expressed by *E. coli* to neutralize the venom of *Loxosceles laeta*. The authors selected six RNA aptamers from a pool of 10^13^ different RNA sequences (with 60 central random nucleotides) using multiple SELEX selection. The RNA aptamers were selected according to their affinities for their targets: three RNAs for SMD-Ll1 and two RNAs for SMD-Ll2. Preselected RNA molecules were reverse-transcribed and subcloned for in vitro transcription purposes. Next, a second round of SELEX selection was performed with these clones, which was based on their inhibitory activity. This allowed the selection of six final aptamers: four with inhibitory activity for SMD-Ll1 and SMD-L12, with cross-reactive inhibition of both targets. As the data on these aptamers (sequence, K_D_, etc.) are too restricted at this stage, they are not presented in Table 2 [64].

## 8. Conclusions

Since antisera were first developed to treat envenomation, research efforts have mostly been devoted to improving their characteristics by making them generally polyvalent using humanized antibodies and working with monoclonal antibodies, and through better understanding of the nature of the toxic venom components. However, little effort has been dedicated, to date, to identifying alternatives to antibodies. The use of aptamers is gaining momentum for a number of reasons: their ease of production; the rapid technologies that have emerged to identify, sequence, and amplify interacting aptamers; and the capability of engineering them. Thus, the pieces of the puzzle are progressively coming together, hinting at a favorable picture in terms of therapeutic use. Several pieces of the puzzle clearly remain missing for aptamers to become credible alternatives to antibodies. Nonetheless, we are getting closer to the big picture. Undoubtedly, the next big step is the proof of concept that aptamers can neutralize a whole snake venom following injection at different time points of intervention using first dedicated animal models. 

## Figures and Tables

**Figure 1 ijms-21-03565-f001:**
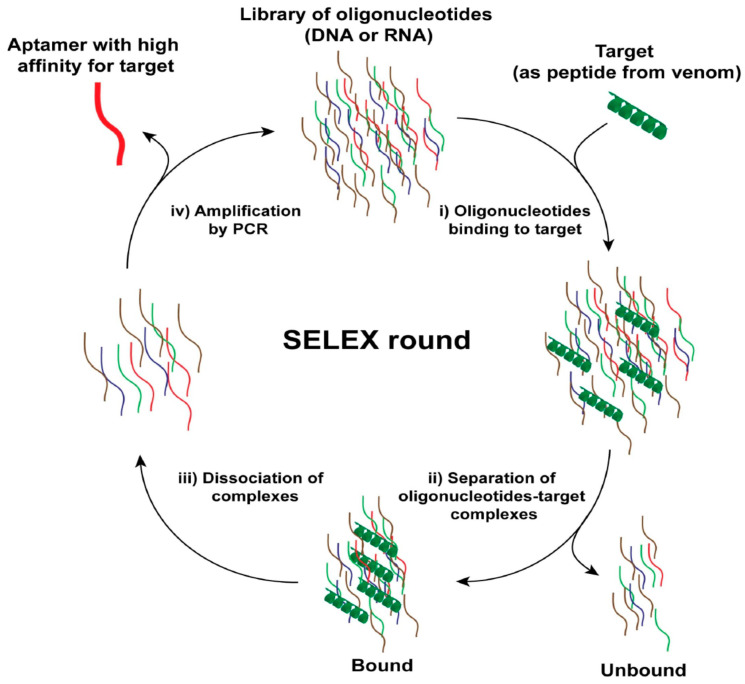
Systematic Evolution of Ligands by Exponential enrichment (SELEX) round for aptamer identification from an oligonucleotide library. The description of the steps includes clockwise: i) incubation of the target antigen with the oligonucleotide library, ii) the removal of the unbound oligonucleotides from the mixture target/library, iii) the dissociation of the complexes, to allow for iv) amplification by PCR of the oligonucleotide hits. PCR amplification of the hits is possible only because the library is designed in such a way that it contains identical sequences at the 5′ and 3′ extremities and a variable core nucleotide sequence.

**Figure 2 ijms-21-03565-f002:**
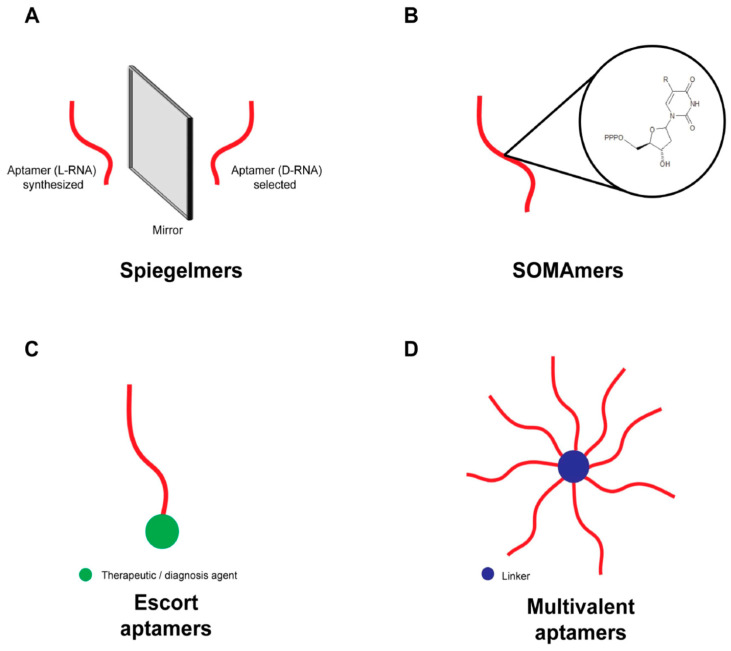
Design of aptamers with the aim to improve function, stability, activity, and/or affinity. (**A**) Spiegelmers. Characterized by the presence of L-ribonucleotides on the RNA sequence to improve stability in biological medium. (**B**) SOMAmers. Sequence containing uridine modified on 5′-position (R) to increase affinity, specificity and stability. (**C**) Escort aptamer. Aptamer linked with a therapeutic or diagnosis agent (i.e., anti-tumor agent) for guiding to its target or conferring a new function. (**D**) Multivalent aptamer. Pool of aptamers combined with a linker to recognize several targets.

**Table 1 ijms-21-03565-t001:** Technological initiatives and progresses in the development of antivenoms. IgG: immunoglobulin G.

Antivenom Class	Target Venoms	Benefits	Disadvantages	References
Monovalent	Snake	Demonstrated clinical efficiency over time	Species dependent – Low paraspecificity – Requires unmistakable species determination	[1,2,3,4,5,6,14,15,16]
Polyvalent	Snake	Improved paraspecificity	Costly development	[11,14]
Based on selected toxins	Spider	Does not require a venom source – Avoids excess needless IgG – Can be polyvalent	Requires i) excellent knowledge of toxic venom components, ii) good toxicovenomic and iii) toxin production capabilities	[12]
Bioinformatics-assisted	Snake	Does not require a venom source - Simplifies the production of the antigens – Can be polyvalent	Requires the knowledge of the venom toxic components and good immunogen potential of chosen epitopes	[30]
Monoclonal antibodies (IgG)	Snake & scorpion	Polyvalence possible, long half-life and low immunogenicity if human origin, few adverse reactions	Limited tissue distribution, large size, complex structure, costly development	[31,32,33,34,35,36,37,38,39]
Fab and F(ab’)_2_ fragments	Snake & scorpion	Polyvalence possible, enlarged tissue distribution and penetration, fewer adverse reactions than IgG	Higher cost of production	[38]
Murine scFv	Snake	Easy to produce, stable, long shelf-life, better tissue distribution and penetration	Shorter half-life *in vivo*, high immunogenicity	[38]
Human scFv	Snake, scorpion and bee	Stable, fewer adverse reactions, large tissue distribution and penetration	Shorter half-life *in vivo*	[38]
Nanobodies	Snake	High affinity and specificity, thermostable, small size (higher tissue penetration), low cost, low immunogenicity	Short half-life (limitation for a longer period time treatment)	[40]
Nanoparticles	Snake	Stability, low cost	Pharmacokinetics issues, low solubility	[41]
Darpins Affibodies Adnectins Avimers Anticalins	Not yet tested	Small size, high stability and solubility, high affinity, cost-effective production, better tissue penetration, low immunogenicity, polyvalence possible, facilitated chemical conjugation, kidney clearance	Short half-life compared to IgG, efficacy for toxin and venom neutralization remains to be demonstrated	[40]
Small molecules	Snake (Varespladib)	High absorbability, low-cost, thermostable, polyvalence	Works on a single class of toxins	[42,43,44]
Phytoantivenom	Snake	Viable alternative to modern medicine and pharmacology	Large extent to the global adverse effects at the clinical level	[45,46,47,48,49,50,51,52,53,54,55,56]
Aptamers	Snake, scorpion, cone snail	Low cost, high stability, long shelf-life, easier chemical conjugations, polyvalence possible	Demonstration lacking for full venom neutralization	[57,58,59,60,61,62,63,64,65]

**Table 2 ijms-21-03565-t002:** List of aptamers developed so far with the aim to neutralize venom toxins. CE: Capillary Electrophoresis; CTX: cardiotoxin; CLPB: cardiotoxin-like protein. All the illustrated aptamers are DNA oligonucleotides. *Aptamers designated bgt1, clones 24 & 51 or α-Tox-FL have identical sequences.

Name	Method	Target	Target Origin	K_D_ (µM)	Sequence of Random Region (5′ to 3′)	References
Clones 24 & 51*	SELEX	α-bungarotoxin	*Bungarus multicinctus*	7.58	GCGAGGTGTTCGAGAGTTAGGGGCGACATGACCAAACGTT	[61]
βB-1	Plate-SELEX	β-bungarotoxin	*Bungarus multicinctus*	0.066	GTTTTCCCCTTGTCGCTTTTGGTTCGTTCTGCCTCTATCT	[65]
βB-20	Plate-SELEX	β-bungarotoxin	*Bungarus multicinctus*	0.084	ATTAGTCATGTTTGTTTGTCTGGCTTTTTGGGTTTGTGCAGTATTATGAAC	[65]
βB-19	Plate-SELEX	β-bungarotoxin	*Bungarus multicinctus*	0.53	TTTGGTGTGGATCCTGAACATTTATATTCTTTCGTTTTTT	[65]
βB-32	Plate-SELEX	β-bungarotoxin	*Bungarus multicinctus*	0.995	GCAATGCACCTTTGTCTCTTATAGTTTATTTTTTGCCTT	[65]
bgt1*	SELEX	α-bungarotoxin	*Bungarus multicinctus*	2.21	GCGAGGTGTTCGAGAGTTAGGGGCGACATGACCAAACGTT	[57]
	SELEX	CTX1, CTX2, CTX3, CTX4, CTX5, CTXN, CLPB	*Naja atra*	2.51, 6.29, 2.25, 8.13, 17.17, 8.85, 7.19		
bgt2	SELEX	α-bungarotoxin	*Bungarus multicinctus*	0.46	AGGGCACAGAGAAGAAGTCGTGGATTTGAATGGTTTTGGT	[57]
	SELEX	CTX3	*Naja atra*	0.26		
bgt3	SELEX	α-bungarotoxin	*Bungarus multicinctus*	0.14	ATCATGTCTTTTCGGGATGGGCAAGAAGGGAAATAATGC	[57]
	SELEX	CTX3	*Naja atra*	1.26		
bgt4	SELEX	α-bungarotoxin	*Bungarus multicinctus*	0.28	AGAAACGTAGCGGTAACTGCTAGAATGCGCCGAGAGAGCG	[57]
	SELEX	CTX3	*Naja atra*	1.17		
α-Tox-FL*	SELEX	crude venom	*Bungarus caeruleus*	0.018	GCGAGGTGTTCGAGAGTTAGGGGCGACATGACCAAACGTT	[57,58,61]
α-Tox-T1	Bioinformatics tools	crude venom	*Bungarus caeruleus*	0.045	GCGAGGTGTTCGAG	[58]
α-Tox-T2	Bioinformatics tools	crude venom	*Bungarus caeruleus*	0.003	AGTTAGGGGCGACATGACCAAACGTT	[58]
D3	CE-SELEX	αC-conotoxin PrXA	*Conus parius*	0.122	ATCGGTCGTATAGGGTCGATTTGGTCGGCA	[59,60]
A5	CE-SELEX	αC-conotoxin PrXA	*Conus parius*	0.184	GTGCAGGTCTATACAGGACAGTCTTCTGAT	[59,60]
D7	CE-SELEX	αC-conotoxin PrXA	*Conus parius*	0.238	TGCAGCATGGGGGATGTGCTCTTCCGCGTG	[59,60]
A4	CE-SELEX	αC-conotoxin PrXA	*Conus parius*	0.246	AATGCTGTTGTTTGAGTATCAATCAGACCG	[59]
B4	CE-SELEX	αC-conotoxin PrXA	*Conus parius*	0.12	TACGCACATACTGTGTACCTTGAATTTATA	[59]
B3	CE-SELEX	αC-conotoxin PrXA	*Conus parius*	> 5	CCGTAGATGCGGGGATGCCAGTCTTGCTTA	[59]

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
