# Peer review of "Diagnostic and Therapeutic Value of Aptamers in Envenomation Cases"

_ijms, 2020, doi:10.3390/ijms21103565_

Round 1
Reviewer 1 Report
Nucleic acid aptamers are a promising alternative to antibodies and other antivenom agents. The authors summarized well the various approaches taken so far to address different envenomation issues, including paraspecificity and synthetic multiepitope. But the reviewer recommends the following points.
- A table summarizing the history of antivenoms together with their pros and cons would benefit the reader.
- The manuscript would be improved a lot if edited by a native speaker of Englsih.
- Please check lines 133, 261, 463, 565, 631 and elsewhere for 'capable of ~ing' and line 26 for 'began'
- Please check lines 257, 503, 519, 534, 548, 555, 565, 581 for unidentifiable letter and line 453 for spelling (affine).
- Please consider changing 'extraction' to 'separation' in Figure 1.
- More description on RNA aptamer would improve the manuscript.
Author Response
Comment:
Nucleic acid aptamers are a promising alternative to antibodies and other antivenom agents. The authors summarized well the various approaches taken so far to address different envenomation issues, including paraspecificity and synthetic multiepitope. But the reviewer recommends the following points.
- A table summarizing the history of antivenoms together with their pros and cons would benefit the reader.
Response:
We thank the reviewer for his/her appreciation of the work. We have now integrated a table that summarizes some of the points described in the manuscript (now Table 1).
Comment:
- The manuscript would be improved a lot if edited by a native speaker of Englsih.
Response:
We understand this concern. We ourselves did our best to correct the manuscript further. We decided also that the manuscript would be corrected by the MDPI English editing service to make sure that it fits the best standards of English language.
Comment:
- Please check lines 133, 261, 463, 565, 631 and elsewhere for 'capable of ~ing' and line 26 for 'began'
- Please check lines 257, 503, 519, 534, 548, 555, 565, 581 for unidentifiable letter and line 453 for spelling (affine).
- Please consider changing 'extraction' to 'separation' in Figure 1.
- More description on RNA aptamer would improve the manuscript.
Response:
We checked these lines and corrected them. We modified Figure 1 accordingly. In the last paragraph, we added a few more lines on RNA aptamer (hopefully the reviewer was making reference to this paragraph).
Reviewer 2 Report
Review is well-written. I do not have major concerns. In my opinion, only minor points need to be addressed before publication.
References given in brackets (lines 59-60, 74, 75, 88, 89, 238-239, 383, 404, 445-446, 475, 478) need to be numbered and added to the list of references.
Some references are missing in lines 190-193, 365-370, 502-504, 586-589. In addition, the most recent and general reviews on conjugated aptamers (discussed in lines 232-236) are missing, please add them:
1) doi: 10.1002/smll.201902248; 2) doi: 10.3390/cancers9120174; 3) doi: 10.3390/molecules22122108.
Please check reference style for lines 694 and 695. In the present form, these references are hard to be found.
Improve english and/or clarify the misleading sentences in the following lines: 226-229, 260-262, 314-315, 384-387, 397-398, 403 (k-,k- is doubled), 419-423, 457-458, 519-521.
Figure 2: Escort aptamers are discussed before multivalent aptamers, reverse figures 2c-2d.
Table 1: Aptamers should be listed following the same order as in the text; replace ‘reference’ with ‘references’; errors on Kd values should be reported for all or for none; for alfa-Tox-T1 and alfa-Tox-T2, I would specify (in method column or in table caption) that they have been selected by means of bioinformatics tools.
Author Response
Comment:
Review is well-written. I do not have major concerns. In my opinion, only minor points need to be addressed before publication.
References given in brackets (lines 59-60, 74, 75, 88, 89, 238-239, 383, 404, 445-446, 475, 478) need to be numbered and added to the list of references.
Response:
With the Covid situation I had issues with my endnote reference base. The database has been reconstituted entirely and we fixed these issues.
Comment:
Some references are missing in lines 190-193, 365-370, 502-504, 586-589. In addition, the most recent and general reviews on conjugated aptamers (discussed in lines 232-236) are missing, please add them:
1) doi: 10.1002/smll.201902248; 2) doi: 10.3390/cancers9120174; 3) doi: 10.3390/molecules22122108.
Please check reference style for lines 694 and 695. In the present form, these references are hard to be found.
Response:
We have added new references where they were missing. We also added the suggested references. Finally, we tried to fix the format of these two references. These are patents that were handled this way by Endnote. We hope we have now an acceptable format.
Comment:
Improve english and/or clarify the misleading sentences in the following lines: 226-229, 260-262, 314-315, 384-387, 397-398, 403 (k-,k- is doubled), 419-423, 457-458, 519-521.
Response:
We have clarified the misleading sentences and also asked the help of the MDPI English editing service.
Comment:
Figure 2: Escort aptamers are discussed before multivalent aptamers, reverse figures 2c-2d.
Response:
We reversed the order of those two panels as requested.
Comment:
Table 1: Aptamers should be listed following the same order as in the text; replace ‘reference’ with ‘references’; errors on Kd values should be reported for all or for none; for alfa-Tox-T1 and alfa-Tox-T2, I would specify (in method column or in table caption) that they have been selected by means of bioinformatics tools.
Response:
We have modified the Table 1 according to these comments. This is now Table 2.